# Prevalence of Hepatitis E Virus (HEV) in Feral and Farmed Wild Boars in Xinjiang, Northwest China

**DOI:** 10.3390/v15010078

**Published:** 2022-12-27

**Authors:** Jian-Yong Wu, Xiao-Xiao Meng, Yu-Rong Wei, Hongduzi Bolati, Eric H. Y. Lau, Xue-Yun Yang

**Affiliations:** 1Xinjiang Key Laboratory of Animal Infectious Diseases, Institute of Veterinary Medicine, Xinjiang Academy of Animal Science, Urumqi 830013, China; 2School of Public Health, Xinjiang Medical University, Urumqi 830016, China; 3School of Public Health, The University of Hong Kong, Hong Kong SAR, China; 4Laboratory of Data Discovery for Health (D24H), Hong Kong Science Park, Hong Kong SAR, China

**Keywords:** Hepatitis E virus, genotype 4, prevalence, phylogenetic analysis, wild boar

## Abstract

Hepatitis E virus (HEV) causes infections in humans and a wide range of animal hosts. Wild boar is an important natural reservoir of HEV genotypes 3–6 (HEV-3–HEV-6), but comparative analysis of HEV infections in both feral and farmed wild boars remains limited. In this study, samples from 599 wild boars were collected during 2017–2020, including 121 feral wild boars (collected 121 fecal, 121 serum, and 89 liver samples) and 478 farmed wild boars (collected 478 fecal and 478 serum samples). The presence of anti-HEV IgG antibodies were detected by the HEV-IgG enzyme-linked immunosorbent assay (ELISA) kit. HEV RNA was detected by reverse transcription polymerase chain reaction (RT-PCR), targeting the partial ORF1 genes from fecal and liver samples, and the obtained genes were further genotyped by phylogenetic analysis. The results showed that 76.2% (95% CI 72.1–79.9) of farmed wild boars tested anti-HEV IgG seropositive, higher than that in feral wild boars (42.1%, 95% CI 33.2–51.5, *p* < 0.001). HEV seropositivity increased with age. Wild boar HEV infection presented a significant geographical difference (*p* < 0.001), but not between sex (*p* = 0.656) and age (*p* = 0.347). HEV RNA in fecal samples was detected in 13 (2.2%, 95% CI 1.2–3.7) out of 599 wild boars: 0.8% (95% CI 0.0–4.5, 1/121) of feral wild boars and 2.5% (95% CI 1.3–4.3, 12/478) of farmed wild boars. Phylogenetic analysis showed that all these viruses belonged to genotype HEV-4, and further grouped into sub-genotypes HEV-4a, HEV-4d, and HEV-4h, of which HEV-4a was first discovered in the wild boar populations in China. Our results suggested that farms could be a setting for amplification of HEV. The risk of HEV zoonotic transmission via rearing and consumption of farmed wild boars should be further assessed.

## 1. Introduction

Hepatitis E virus (HEV) is known as a hepatotropic virus with the capability of causing acute viral hepatitis in humans, and chronic hepatitis in immunocompromised patients [1,2,3]. HEV infection presents worldwide distribution, and is mainly found in low- and middle-income countries [4]. HEV is primarily transmitted through the fecal–oral route, i.e., via contaminated water and food, and through occupational exposure to animal [5,6]. About 20 million HEV infections are reported annually, leading to 3.3 million persons developing the symptoms of hepatitis and 44,000 deaths [7]. Globally, HEV remains a significant public health problem, with roughly 1 out of 8 people ever been infected [8].

Hepatitis E virus (HEV) belongs to the family *Hepeviridae* and subfamily *Orthohepevirinae,* which can proliferate in a wide range of susceptible hosts, including humans, pigs, rabbits, boars, rats, red deer, birds, and others [9,10,11,12]. The subfamily *Orthohepevirus* is further classified into 4 genera, including *Paslahepevirus*, *Avihepevirus*, *Rocahepevirus*, and *Chirohepevirus*, and genera *Paslahepevirus* and *Rocahepevirus* were shown to have zoonotic potential [12,13,14]. Species *Paslahepevirus balayani* belongs to the genus *Paslahepevirus*, and have been assigned to 8 difference genotypes (HEV-1–HEV-8) infecting humans, pigs, rabbits, camels, etc. [15]. Genotypes HEV-1 and HEV-2 are mainly found in humans [16], while HEV-3 and HEV-4 are considered as common zoonotic viruses that infect humans and several other mammal species, including wild boars [17]. HEV-5 and HEV-6 were only found in wild boars [18], HEV-7 in dromedary camels and humans [19,20], and HEV-8 could infect Bactrian camels, rabbit, and *Cynomolgus Macaques* that possessed zoonotic potential [21,22,23].

In China, HEV was endemic in the general human population, with a reported seroprevalence of 23.5% [24]. A serious outbreak of HEV-1 occurred in Xinjiang, a northwest province in China, where a total of 119,280 cases and 705 deaths were reported during 1986–1988 [25]. A recent investigation showed that the HEV incidence increased to 2.0 per 100,000 in 2019, and the dominant genotype shifted from HEV-1 to zoonotic genotype HEV-4 [26,27]. A small foodborne outbreak of zoonotic HEV-4 occurred in Shandong province, China, in an employee cafeteria [28]. HEV-3 and HEV-4 were also dominant in humans in Shanghai, China [14]. Wild boars can be infected with zoonotic genotypes HEV-3 and HEV-4 [18], which may become natural reservoirs of the viruses and transmit to animals in the farming settings and even humans [19]. The HEV seroprevalence in wild boars ranged from 5% to 59% in European countries, 0 to 71% in Asian countries, and was reported to be 3% in the USA [18]. In China, the findings on the prevalence and molecular characteristics of HEV infection in wild boars are still limited, even in farmed wild boars for meat consumption. A previous study showed that antibodies to HEV were detected in 24.5% of the farmed wild boars [29]. Another study showed that 28.2% of the wild boars were seropositive, among which HEV-4d, HEV-4g, and HEV-4h were identified [30]. Circulation of HEV in wild boars poses a zoonotic risk, and severe HEV infections after consumption of uncooked wild boar livers have been reported [31]. Compared to farmed wild boars, the seroprevalence of *Toxoplasma gondii* infections was found to be higher among feral wild boars [32]. However, the comparison of HEV infections between feral and farmed wild boars is not well-studied. 

In this study, we collected samples from both feral and farmed wild boars in Xinjiang, Northwest China, during 2017–2020. China banned the breeding of feral wild boars in 2019. In our study, the farmed wild boars are hybrids of farmed pigs and wild boars or their descendants but preserve the appearance of feral wild boars. We compared the serologic and molecular detection of HEV infection in both feral and farmed wild boars.

## 2. Materials and Methods

### 2.1. Sample Collection

Xinjiang Uygur Autonomous Region (34°25′ N–48°10′ N, 73°40′ E–96°18′ E) is the largest province in China, comprising 14 prefectures with a total land area of 1,664,900 km^2^. Feral and farmed wild boars were hunted and/or sampled with the consent of the Xinjiang or Prefecture’s Forestry and Grassland Bureau and samples were processed according to the Law of The People’s Republic of China on the Protection of Wildlife. The age of feral wild boars was roughly estimated based on body weight, tusks, and neck mane, and the exact age of farmed wild boars was provided by the farmers. Blood, feces, and liver samples were collected after hunting and dissecting from feral wild boars. For farmed wild boars, blood samples were collected from the anterior vena cava, and feces were collected from the rectum. In total, we collected 599 serum, 599 fecal, and 89 liver samples from wild boars in 11 prefectures during 2017–2020, including 478 from farmed wild boars (farmed offspring of male wild boars and female domestic pigs preserving the appearance of wild boar) and 121 from feral wild boars. Of which, we collected the 121 feral wild boars (including 121 fecal, 121 serum, and 89 liver samples) in 7 prefectures during 2017–2020. Fecal, liver, and serum samples were collected from all feral wild boars, except that we failed to collect liver samples from 32 feral boars. In addition, we collected 478 fecal and 478 serum samples from all farmed live wild boars in 33 farms over 9 prefectures in 2019 (Figure 1, Table 1). Liver samples were not collected from farmed live wild boars. All samples were transported on dry ice and stored at −80 °C until testing. All serum samples were tested for anti-HEV antibodies, and all fecal and liver samples were tested for HEV RNA. 

### 2.2. Serological Testing

Serum samples were tested for anti-HEV antibodies using commercially available HEV ELISA testing kits (Wantai BioPharm, Beijing, China). Since the kit is for human use, we replaced the secondary antibody with protein G horseradish peroxidase (1:10,000 dilution) (Thermo Fisher Scientific, Eugene, OR, USA) as a conjugate, following the same procedure as described by Elemi et al. [33]. Other experimental procedures were carried out as instructed by the manufacturer. Ten sera were identified by Western blotting for the absence of antibodies against the HEV ORF2 protein (Abcam, Cambridge, UK) and then pooled with equal volume as negative control sera, which had OD_450nm_ values ranging from 0.030 to 0.054 detected by ELISA. The cutoff value was calculated after negative serum as the mean OD_450nm_ of the negative controls +0.16 according to the instructions, above which it was considered positive.

### 2.3. RNA Extraction

Total RNA was extracted with the TIAamp Viral DNA/RNA Mini kit (Tiangen Biotech, Beijing, China) following the manufacturer’s instructions using 200 mg of liver or fecal samples. 

### 2.4. RT-PCR and Sequencing

The detection of HEV RNA was conducted by PCR targeting the partial ORF1 genes, as previously described [34,35]. PCR was carried out using the 2 × PCR Mastermix (Promega, Madison, WI, USA) using 4 μL of extracted RNA and 16 μL of master mix according to the manufacturer’s recommendations. First-round RT-PCRs were carried out by using a protocol with reverse transcription at 42 °C for 60 min and subsequent PCR at 95 °C for 4 min, 30 cycles of denaturation at 94 °C for 30 s, annealing at 50 °C for 30 s, and extension at 72 °C for 35 s. Second-round reactions used the same cycling protocol but without the RT step. Amplicons of the expected size (338 bp in the second round) were visualized on 2.0% agarose gels with ethidium bromide staining. All PCR products were sequenced using the Sanger sequencing method (Sangon Biotech, Shanghai, China). 

### 2.5. Phylogenetic Analysis

Multiple sequence alignment was performed using MAFFT version 7 (https://mafft.cbrc.jp/alignment/server/, accessed on 4 September 2022) [36]. The alignment was manually checked and end-trimmed to match to the obtained sequence. The phylogenetic tree was interfered with Mega 11 version 12.0.0 (http://www.megasoftware.net/, accessed on 4 September 2022) [37]. The final multiple sequence alignment was used for maximum likelihood (ML) phylogenetic analysis with TN93 + G as the best-fit model of nucleotide substitution and 1000 bootstrap resampling by using Mega 11.

### 2.6. Statistical Analysis

We present the seroprevalence of HEV by prefecture, types of wild boars, and sex. Chi-squared or Fisher’s Exact tests were used to compare HEV seropositivity between prefectures, farming types, and sex. Clopper–Pearson confidence intervals were constructed for proportions. A *p*-value less than 0.05 was considered statistically significant. Analyses were carried out using the SPSS 20.0 software package (IBM Corp, Armonk, NY, USA).

## 3. Results

### 3.1. Overall Seroprevalance in Wild Boar Populations

Of the 599 wild boars tested, 415 (69.3%, 95% confidence interval (CI) 65.4%–73.0) were seropositive for HEV. HEV seropositivity varied across prefectures, ranging from 0 to 100%. The highest HEV seropositivity was found in Aksu (100.0%, 95% CI 47.8–100.0), followed by Tacheng (88.5%, 95% CI 80.7–93.9), and the lowest in Hami (23.8%, 95% CI 8.2–47.2), Altay (9.1%, 95% CI 0.0–41.2), and Hotan (0.0%, 95% CI 0.0–84.2) (Table 2). There was no significant difference in HEV seropositivity between sex (*p* = 0.656) and age (*p* = 0.347). The seroprevalence of farmed wild boars (76.2%, 95% CI 72.1–79.9) was significantly higher than that of feral wild boars (42.1%, 95% CI 33.2–51.5, χ^2^ = 7.825, *p* < 0.001).

### 3.2. HEV Seroprevalence in Feral and Farmed Wild Boar Populations

Seropositive feral wild boars were found in five out of seven prefectures (Table 3). Both feral and farmed wild boars showed clear age patterns in HEV seroprevalence (*p* = 0.001 and < 0.001, respectively). Comparing wild boars 12 months or younger, farmed wild boars had a significantly higher HEV seroprevalence (75.1% versus 21.6% *p* < 0.001). High HEV seropositivity was found in Urumqi (68.0%, 95% CI 53.3–80.5) and low HEV seropositivity in Ili (10.6%, 95% CI 3.5–23.1). Seropositive farmed wild boars were found from all nine prefectures (Table 3). High HEV seropositivity was found in Aksu (100.0%, 95% CI 47.8–100.0), Bayingolin (73.7%, 95% CI 48.8–90.9), Tacheng (89.8%, 95% CI 82.0–95.0), and Urumqi (91.5%, 95% CI 79.6–97.6). Low HEV seropositivity was found in Altay (12.5%, 95% CI 0.0–52.7) and Hami (23.8%, 95% CI 8.2–47.2). 

### 3.3. Molecular Detection of HEV RNA and Age-specific HEV RNA Positivity

HEV RNA was detected in 19 out of 599 wild boars (3.2%, 95% CI 1.9–4.9), all from seropositive wild boars. Of which, 7 out of 89 (7.9%, 95% CI 3.2–15.5) liver samples from feral wild boars were HEV RNA-positive, of which one feral wild boar tested HEV RNA-positive in both fecal and liver samples, and 2.5% of farmed wild boars (12/478, 95% CI 1.3–4.3) were detected as HEV RNA-positive from fecal samples. We could not identify a significant age difference of HEV RNA positivity from fecal samples in feral and farmed wild boars (*p* = 1.000 and 0.701, respectively), and liver samples had a much higher HEV RNA-positive rate (Appendix A). Comparing wild boars 12 months or younger, farmed wild boars had a higher HEV seroprevalence (2.6% versus 2.0%, *p* = 1.000), though the difference did not reach statistical significance. Sub-genotypes HEV-4a and HEV-4d were most commonly identified from feral and farmed wild boars, respectively (Table 4). 

### 3.4. Phylogenetic Tree

After sequencing of the positive PCR amplicons obtained from partial ORF1 genes, all the HEVs were clustered into genotype 4, and further grouped into sub-genotypes HEV-4a, HEV-4d, and HEV-4h (Figure 2). Sub-genotype HEV-4a was firstly identified from wild boars in China, in two neighboring prefectures (Changji and Urumqi), and from a farmed wild boar in Urumqi (Table 4). HEV from the farmed wild boars included sub-genotypes HEV-4a, HEV-4d, and HEV-4h, while HEV in feral wild boars included sub-genotypes HEV-4a and HEV-4h. The results indicated that sub-genotypes HEV-4a and HEV-4h circulated in both feral and farmed wild boars in Xinjiang, Northwest China, though we could not rule out circulation of HEV-4d in farmed wild boars. 

## 4. Discussion

Wild animals are the natural reservoirs for many pathogens, some of which are closely linked to human health [38,39]. Wild boars harbor more than 20 pathogens, including zoonotic pathogens, such as HEV, *Mycobacterium tuberculosis* complex, *Brucella suis*, *Salmonella enterica*, *Yersinia* spp., and others [40,41]. Humans and other farmed animals are exposed to pathogens circulating among farmed wild boars in the farm setting. Thus, comparing the difference of HEV infection between farmed and feral wild boars would be important to understand the transmission risk in different settings. In this study, the HEV seroprevalence was 42.1% in feral wild boars, and 76.2% in farmed wild boars. The seroprevalence of HEV in the farmed wild boar population was comparable to earlier reports in Japan, where 71.4% (10/14) of farmed wild boars were seropositive for HEV [42], and comparable to a study in China, where 65% (ranging from 35% to 92%) of domestic pigs were seropositive for HEV [43], and the HEV seroprevalence for adult domestic pigs was reported to reach >90% in France [44]. We also observed a higher HEV RNA positivity among farmed wild boars 12 months or younger compared to feral wild boars (2.6% versus 2.0%), though the difference did not reach statistical significance. This is in line with the observed higher seroprevalence among farmed wild boars, indicating a higher infection risk in farms. This suggested that the environmental conditions of farmed wild boars, such as limited space or poor sanitary conditions, may better facilitate amplification of HEV than in the natural environments. 

Wild boars can be infected with genotypes HEV-3–HEV-6 in other countries [45], and our study first identified sub-genotype HEV-4a from wild boars in China. Together with Gong et al. [30], our results indicated that only HEV-4 (e.g., HEV-4a, HEV-4d, HEV-4g, and HEV-4h) circulated among wild boars in China. The high seroprevalence suggested that feral and farmed wild boars are zoonotic HEV reservoirs. In fact, HEV genotype 4 was mostly reported in Asia [46]. In future studies, the genetic association of HEV between humans and wild boars should be investigated, since domestic pig and wild meat consumption play an importance role for human HEV infection [47,48]. 

A previous study [30] reported 2.7% (9/331) HEV RNA positivity from liver samples of wild boars in 25 provinces of China and 8.3% (1/12) in Xinjiang, consistent with our results (7.9%, 7/89). In our study, the HEV RNA positivity rate from liver samples was more than 10-fold higher than that in fecal samples (Appendix A), probably due to its hepatotropic characteristics that HEV RNA concentration in liver and bile were 10-fold higher than that in fecal and other samples [49]. However, HEV RNA positivity from fecal samples allows a fair comparison between feral and farmed wild boars, and a similar age distribution can be observed, with a higher HEV RNA positivity among 7–12-month-old feral wild boars (2.3%) and 7–9-month-old farmed wild boars (4.0%), respectively (Appendix A). This study also suggested some geographical variation in HEV prevalence across provinces in China. We also observed substantial geographic variation in HEV prevalence among feral wild boars within prefectures, while high HEV prevalence was consistently observed in farms, although we did not observe a clear connection between specific sub-genotypes among farmed and feral wild boars from virologic data (Table 4). Further HEV molecular studies in farmed wild boars are needed to elucidate whether HEV originates from feral wild boars, domestic pigs, or occupational workers.

Our study had some limitations. First, there were a limited number of samples where the HEV genotype was determined to draw conclusions about differences in HEV RNA positivity as well as HEV genotype distribution between farmed and feral wild boars. Second, since the focus of our study was to compare the seroprevalence and genotype differences between feral and farmed wild boars, we also did not measure the viral load of HEV RNA-positive samples.

## 5. Conclusions

Our findings confirmed that wild boars in Xinjiang, Northwest China, had higher prevalence of HEV infections, compared to other provinces. Substantial regional differences in HEV seroprevalence were observed, and farmed wild boars had a significantly higher infection risk than that of feral wild boars, indicating farms as a setting for viral amplification. Sub-genotype HEV-4a was first identified from the wild boar populations in China. HEV-4a, HEV-4d, and HEV-4h were the main circulating sub-genotypes, and HEV-4a and HEV-4h had infected both feral and farmed wild boars. Further studies may elucidate the risk and route of HEV cross-species and zoonotic transmission.

## Figures and Tables

**Figure 1 viruses-15-00078-f001:**
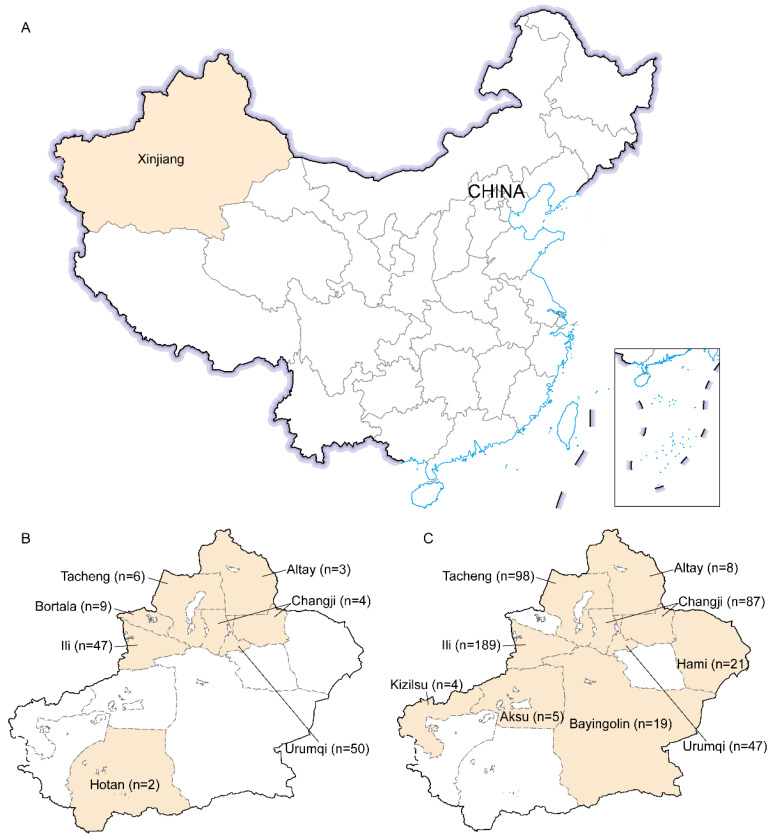
A simplified map of Xinjiang (Northwest China), where Hepatitis E virus strains were collected. The geographical locations of the strains detected in the study are marked in yellow. (**A**) the location of Xinjiang in China. (**B**) The prefectures where samples were collected from feral wild boars. (**C**) The prefectures where samples were collected from farmed wild boars.

**Figure 2 viruses-15-00078-f002:**
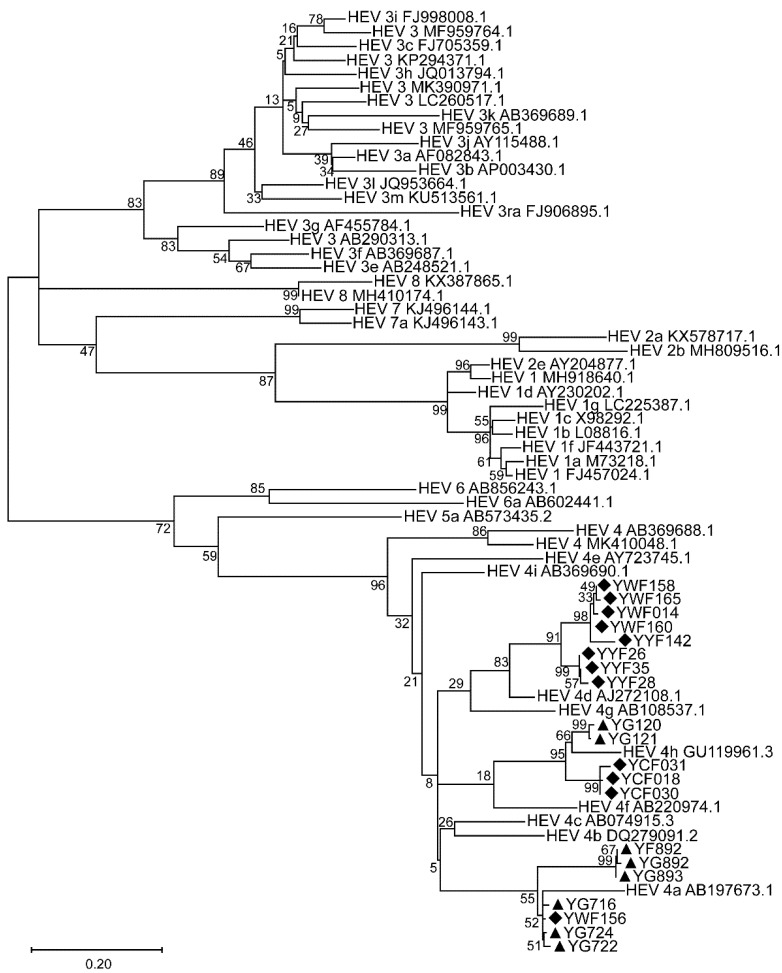
Phylogenetic tree of partial ORF1 of HEV from feral and farmed wild boars. The evolutionary history was inferred by using the Maximum Likelihood method based on the TN93 + G model, a partial nucleotide sequence of the ORF1 region, and reported HEV sequences in GenBank as references. One thousand resamplings of the data were used to calculate percentages (values along branches) of tree branches obtained. Black triangles indicate HEV strains from feral wild boars, and black diamonds indicate HEV strains from farmed wild boars.

**Table 1 viruses-15-00078-t001:** Number of samples collected in this study.

Sample Type	No. of Samples Collected
Feral wild boars	
Serum	121
Feces	121
Liver	89
Farmed wild boars	
Serum	478
Feces	478
Total	1287

**Table 2 viruses-15-00078-t002:** Overall seroprevalence of HEV in feral and farmed wild boar populations.

Variable	No. Tested	No. Positive	Seroprevalence, %	95% CI	*p*-Value
Prefecture *					<0.001
Aksu	5	5	100.0	(47.8–100.0)	
Altay	11	1	9.1	(0.0–41.3)	
Bayingolin	19	14	73.7	(48.8–90.9)	
Bortala	9	4	44.4	(13.7–78.8)	
Changji	91	65	71.4	(61.0–80.4)	
Hami	21	5	23.8	(8.2–47.2)	
Hotan	2	0	0.0	(0.0–84.2)	
Kizilsu	4	3	75.0	(19.4–99.4)	
Tacheng	104	92	88.5	(80.7–93.9)	
Urumqi	97	77	79.4	(70.0–86.9)	
Ili	236	149	63.1	(56.6–69.3)	
Type					<0.001
Feral	121	51	42.1	(33.2–51.5)	
Farmed	478	364	76.2	(72.1–79.9)	
Sex					0.656
Female	267	188	70.4	(64.5–75.8)	
Male	332	227	68.4	(63.1–73.3)	
Age (months)					0.347
0–6	268	187	69.8	(63.9–75.2)	
7–12	245	174	71.0	(64.9–76.6)	
>12	86	54	62.8	(51.7–73.0)	
Total	599	415	69.3	(65.4–73.0)	

Abbreviations: CI, confidence interval. *, indicated the city-level administrative region.

**Table 3 viruses-15-00078-t003:** HEV seroprevalence in feral and farmed wild boar populations.

	Feral Wild Boars						Farmed Wild Boars				
	No. Tested	No. Pos	%	95% CI	*p*-Value		No. Tested	No. Pos	%	95% CI	*p*-Value
Age (months)					0.001	Age (months)					<0.001
0–6	8	1	12.5	(0.0–52.7)		0–3	66	30	45.5	(33.1–58.2)	
7–12	43	10	23.3	(11.8–38.6)		4–6	194	156	80.4	(74.1–85.8)	
13–24	10	4	40.0	(12.1–73.8)		7–9	100	78	78.0	(68.6–85.7)	
25–36	25	13	52.0	(31.3–72.2)		10–12	102	86	84.3	(75.8–90.8)	
>36	35	23	65.7	(47.8–80.9)		>12	16	14	87.5	(61.7–98.4)	
Prefecture					<0.001						<0.001
Aksu	-	-	-	-			5	5	100.0	(47.8–100.0)	
Altay	3	0	0.0	(0.0–70.8)			8	1	12.5	(0.0–52.7)	
Bayingolin	-	-	-	-			19	14	73.7	(48.8–90.9)	
Bortala	9	4	44.4	(13.7–78.8)			-	-	-	-	
Changji	4	4	100.0	(39.8–100.0)			87	61	70.1	(59.4–79.5)	
Hami	-	-	-	-			21	5	23.8	(8.2–47.2)	
Hotan	2	0	0.0	(0.0–84.2)			-	-	-	-	
Kizilsu	-	-	-	-			4	3	75.0	(19.4–99.4)	
Tacheng	6	4	66.7	(22.3–95.7)			98	88	89.8	(82.0–95.0)	
Urumqi	50	34	68.0	(53.3–80.5)			47	43	91.5	(79.6–97.6)	
Ili	47	5	10.6	(3.5–23.1)			189	144	76.2	(69.5–82.1)	

Abbreviations: CI, confidence interval; pos, positive.

**Table 4 viruses-15-00078-t004:** Summary of HEV RNA-positive samples *.

Sample No.	Sample Type	Sub-Genotype	Prefecture	Accession No.
Feral wild boars (*n* = 8)				
YF892 *	feces	HEV-4a	Changji	OP577963
YG120	liver	HEV-4h	Ili	OP577964
YG121	liver	HEV-4h	Ili	OP577965
YG716	liver	HEV-4a	Urumqi	OP577966
YG722	liver	HEV-4a	Urumqi	OP577967
YG724	liver	HEV-4a	Urumqi	OP577968
YG892 *	liver	HEV-4a	Changji	OP577969
YG893	liver	HEV-4a	Changji	OP577970
Farmed wild boars (*n* = 12)				
YCF018	feces	HEV-4h	Changji	OP577960
YCF030	feces	HEV-4h	Changji	OP577961
YCF031	feces	HEV-4h	Changji	OP577962
YWF014	feces	HEV-4d	Urumqi	OP577971
YWF156	feces	HEV-4a	Urumqi	OP577972
YWF158	feces	HEV-4d	Urumqi	OP577973
YWF160	feces	HEV-4d	Urumqi	OP577974
YWF165	feces	HEV-4d	Urumqi	OP577975
YYF26	feces	HEV-4d	Ili	OP577977
YYF28	feces	HEV-4d	Ili	OP577978
YYF35	feces	HEV-4d	Ili	OP577979
YYF142	feces	HEV-4d	Ili	OP577976

* Indicates the samples were collected from the same wild boars.

## Data Availability

The sequences of HEV were deposited at GenBank under the accession codes OP577960–OP577979.

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
