# Peer review of "Prevalence of Hepatitis E Virus (HEV) in Feral and Farmed Wild Boars in Xinjiang, Northwest China"

_viruses, 2022, doi:10.3390/v15010078_

Round 1

Reviewer 1 Report

The paper by Jian-Yong Wu et al. reports the virological and serological prevalence of HEV in feral and farmed wild boars in Northwest China. The paper reports the first discovery of HEV-4a subtype in China.

The paper is scientifically correct, well written, and the results are interesting.

However, as far as the content is concerned, I suggest the authors to take into account the following comments.

Major concerns

Introduction. I suggest the authors briefly report the results of the evaluation of seroprevalence and virological prevalence of HEV in other countries and continents, especially those in WBs.

 L. 14. Specify for which HEV genotypes wild boars are an important reservoir.

 L. 55-59. It must be clarified whether the reported prevalence (23.5%) is a seroprevalence (as I believe) or a virological prevalence. Furthermore, the genotype of HEV responsible for the serious outbreak occurred in Xinjiang needs to be specified. I also suggest that the Authors specify the number of human cases/year of hepatitis E reported in China caused by zoonotic genotypes (possibly specifying the subtype).

 L. 64. The paper cited by the authors reports the results of a serological evaluation. Writing that "24.5% of the farmed wild boars were infected with HEV" is incorrect. The sentence needs to be rewritten.

L. 68-69. See previous comment. Again, this is an evaluation of seroprevalence.

Paragraph Sample collection. This paragraph should provide clear information on how and when/where samples are collected. For example, it is not clear when and where the samples of feral and farmed WB were taken: after hunting, at the slaughterhouse, etc.).

For the correct interpretation of the successful results, it should be made explicit already in this paragraph that no information on the age of the animals is available. However, it would be useful to provide some rough indication: I do not believe, for example, that the hunted animals are young WB.

L. 82 and L. 87. Clearly specify: a) whether serum and feces samples are from the same animal (as I assume); b) specify the number of animals tested (total, farmed WB, feral WB, male female); c) specify how and when serum, feces, and liver samples were collected (at the slaughterhouse?); d) specify why liver samples were not collected from farmed WB.

The following should also be specified: a) the number of WBs/km2 in different prefectures and the characteristics of the area; and, b) the number of farms where samples were taken.

In my opinion, the most critical aspect is the choice of the number of samples to be examined for each prefecture (see also later comments). The range is very wide (min 2, max 236). Unless there are substantial differences between prefectures, this large variability does not allow for robust comparisons of different prevalence values between prefectures.

In the Results paragraph, reference is made to a Seasonal pattern. There is no mention of this in the Sample collection paragraph.

L. 103-104. I cannot find the results of the analysis by western blotting.

Paragraph Statistical analysis. Provide information on the method of calculating CIs (e.g., Asymptotic (Wald) method based on a normal approximation; Binomial (Clopper-Pearson) "exact" method based on the beta distribution).

L. 132. Specify which groups are compared.

L. 139-141. See my previous comment (Sample collection).

L. 142. Provide the p-value.

L. 149. Results for farmed WB are also given in the subsection.

L. 151-152 and L. 154-159. See my previous comment (Sample collection).

L. 174-177. What is written here is likely caused by a much larger number of samples taken in Winter. Assuming Expected minimum prevalence of 2% you would need to examine about 150 samples to get 95% probability that at least one is positive. I suggest deleting the paragraph and related table.

L. 213-215. Considering that the farming conditions of farmed WB and pigs are (probably) quite similar, it would be useful to provide some seroprevalence values in domestic pigs in China and/or other countries.

L. 226-228. Considering my previous comments, I suggest the Authors delete this part of the text or at least, specify that the size of the sampling does not allow to say with certainty that there are real differences in prevalence between prefectures.

L. 231-234. Again. See my previous comment and other comments.

L. 241-246. I suggest the authors add the limitations highlighted earlier in my comments.

L. 248-250. See my previous comment (L. 226-228).

Minor concerns

L. 2. Suggest adding "(HEV)" after "Hepatitis E Virus"

L.21 and also elsewhere. Delete "%" after the upper limit of the confidence interval. e.g., "76.2% (95% CI 72.1-79.9)" in place of "76.2% (95% CI 72.1-79.9%)."

L. 22. Replace "p-value < 0.0001" with "p < 0.0001".

L. 24. Replace "p=0.656" with "p=0.656".

L. 77. Delete "and testing". In this paragraph, only the sample collection is described.

L. 98. Replace "ELISA" with something like "serological testing."

L. 113. Papers by Drexler et al., 2012 and Wang et al., 2018 are not shown in the References. Use the journal style [##].

Author Response

The authors wish to thank  the reviewer for your astute observations and constructive comments which help to significantly improve the manuscript.

Reviewer 2 Report

Jian-Yong Wu et al report the molecular surveillance for HEV in Xinjiang Uygur Autonomous Region of feral wild boar and farmed wild boars. Overall, the methods are convincing and the results are interesting. However, I have a few minor concerns that should be addressed:

Line 42: epidemiological data dating back to 2015 it would be appropriate to include more updated data

line 56: reported prevalence of 23,5% was dated 2006. Exist the latest data?

Reference 24: the reference is impossible to consult because it is written in the Chinese language

Author Response

Comments and Suggestions for Authors

Jian-Yong Wu et al report the molecular surveillance for HEV in Xinjiang Uygur Autonomous Region of feral wild boar and farmed wild boars. Overall, the methods are convincing and the results are interesting. However, I have a few minor concerns that should be addressed:

Response: Thank you for your comments.

Line 42: epidemiological data dating back to 2015 it would be appropriate to include more updated data.

Response: Thanks for your comment. However, we were not able to find more updated global data. The World Health Organization still cited this data now (https://www.who.int/news-room/fact-sheets/detail/hepatitis-e), probably the most updated data available.

line 56: reported prevalence of 23.5% was dated 2006. Exist the latest data?

Response: Thanks for your comment. Unfortunately, no more national survey on HEV seroprevalence were conducted in China since 2006.

Reference 24: the reference is impossible to consult because it is written in the Chinese language

Response: We apologize for the inconvenience. The following reference is now replacing the reference in Chinese.

       (page 11) Nelson, K.E.; Heaney, C.D.; Kmush, B.L. The Epidemiology and Prevention of Hepatitis E Virus Infection. Current Epidemiology Reports 2017, 4, 186-198, doi:10.1007/s40471-017-0109-9.

Reviewer 3 Report

The present study was conducted to compare the seroprevalence and genotype differences between feral and farmed wild boars in Xinjiang, northwest China.

Overall, the manuscript provides interesting information regarding HEV infection in wild boars in Xinjiang, China. However, there are several serious concerns that need to be addressed and the present manuscript should be rewritten extensively.

Comments:

1.        Taxonomical classification of family Hepeviridae has recently been updated and the virus family is divided into two subfamilies Orthohepevirinae with four genera (Paslahepevirus, Avihepevirus, Rocahepevirus, and Chirohepevirus) and Parahepevirinae with one genus (Piscihepevirus) (Purdy et al., J Gen Virol. 2022; 103:001778). Therefore, the descriptions in the Introduction section (lines 44-49) should be updated.

2.        It is well known that age of domestic pigs and farmed wild boars significantly affects the prevalence of HEV antibodies and HEV RNA (Vet Microbiol 132:19-28, 2008; Animals 12:272, 2022; Acta Tropica 192:87-90, 2019). Therefore, age information of farmed wild boars is important for interpretation of the results in the present study. The lack of age information is fatal in this paper and scientifically sound comparison of the prevalence of HEV antibodies and HEV RNA between feral and farmed wild boars is impossible. If age information is unavailable, at least body weight-dependent prevalence of HEV antibodies and HEV RNA should be indicated for feral and farmed wild boars and discussed accordingly.

3.        Although the prevalence of anti-HEV was significantly higher in farmed wild boars than in feral wild boars (76.2% vs. 42.1%, p <0.0001), that of HEV RNA was lower in farmed wild boars than in feral wild boars (2.5% vs. 7.9%). This inconsistent result between feral and farmed wild boars needs to be discussed in relation to possible effects of sampling vias.

4.        The number of samples whose HEV genotype was determined in the present study was too small to draw a plausible conclusion regarding the difference of HEV genotype distribution between feral and farmed wild boars (n=8 vs. n=12 in Table 4). This should also be mentioned in the study limitations (lines 241-246).

5.        The genotype distribution was not consistent between feral and farmed wild boars within each prefecture. In detail, in Changji prefecture, HEV-4a was found in feral wild boars, while HEV-4h was found in farmed wild boars. In Ili prefecture, HEV-4h was detected from feral wild boars, while HEV-4d was detected from farmed wild boars. In addition, in Urumqi prefecture, HEV-4a was recovered from feral wild boars, while HEV-4d was predominant in farmed wild boars. These results do not support HEV spillover events from feral wild boars to the farming settings and should be discussed properly.

6.        The sentences on lines 231-234 are confusing. Although HEV RNA was detected exclusively in winter in feral and farmed wild boars, HEV seroprevalence was rather low in winter in both feral and farmed wild boars (Table 5). This seasonal difference in HEV RNA positivity was not statistically significant. These unconvincing statements on lines 173-177 and 231-234 and Table 5 should be removed.

7.        Mega 11 (line 127) needs reference(s). The sentence on lines 222-223 needs relevant reference(s).

8.        Tables 3 and 5 should include total no. tested, no. positive, prevalence and 95% CI as in Table 2.

9.        Figure 2: As for construction of phylogenetic tree, it is mentioned in the methods section (2.5 subsection) that ML phylogenetic analysis with TN93 + G as the best-fit model of nucleotide substitution. However, the legend to this figure indicates that ML method based on the Tamura-Nei model was used (lines197-198). The descriptions regarding the methods used should be consistent throughout the manuscript.

10.    Typographical and grammatical errors are scattered throughout the manuscript. Singular and plural notations are improperly mixed in the text, tables and legends to figures, such as wild boar and wild boars, gene and genes, and strain and strains. What does “approximately 338 bp” (line 119) mean? The sentence on lines 192-193 has no verb.

Author Response

(The authors gave the same response as above.)

Round 2

Reviewer 1 Report

Dear Authors,
I thank you for accepting my suggestions. The quality of the manuscript has significantly improved.
Therefore, I believe that the manuscript can be accepted in the present form.

Author Response

Thank you for your constructive comments which help to significantly improve the manuscript.

Reviewer 3 Report

The manuscript has been generally improved in accordance with my previous comments. However, there are still several concerns that need to be addressed.

Comments:

1.    The positive rate of HEV RNA between feral and farmed wild boars should be compared with samples common to both. Therefore, the sentence in Abstract (lines 26-28) should be changed to “HEV RNA in fecal samples was detected in 13 (2.2%) out of 599 wild boars: 0.8% (1/121) feral wild boars and 2.5% (12/478) farmed wild boars”.

2.    In relation to item 5 in my previous comments, statements regarding “spillover events from feral wild boars to the farm settings or even to humans” should also be deleted from abstract (line 31-32).

3.    Line 241-242: age information of wild boars has been made available in the revised manuscript. The sentence “age information of wild boars was not available for further interpretation of HEV serology” should be removed.

4.    The authors’ response to item 3 in my previous comment (about significant sampling vias) should be described in the Discussion section.

5.    The authors found significantly higher seroprevalence of HEV in farmed wild boars than in feral wild boars (Table 3). However, there were no statistically significant differences between feral wild boars and farmed wild boars in relation to the overall and age-specific prevalence of fecal HEV RNA in the age groups of 0-6, 7-12, and >12 months (Supplementary Table S1). The observed different tendency of the seroprevalence and positivity of fecal HEV RNA between feral and farmed wild boars should be discussed convincingly to support the authors’ claim that farms could be a setting for amplification of HEV.

Author Response

Comments:

  1. The positive rate of HEV RNA between feral and farmed wild boars should be compared with samples common to both. Therefore, the sentence in Abstract (lines 26-28) should be changed to “HEV RNA in fecal samples was detected in 13 (2.2%) out of 599 wild boars: 0.8% (1/121) feral wild boars and 2.5% (12/478) farmed wild boars”.

Response: Thank you for your comment. We had revised the sentence and added the 95% confidence interval accordingly:

“HEV RNA in fecal samples was detected in 13 (2.2%, 95% CI 1.2–3.7) out of 599 wild boars: 0.8% (95% CI 0.0–4.5, 1/121) feral wild boars and 2.5% (95% CI 1.3–4.3, 12/478) farmed wild boars”

  1. In relation to item 5 in my previous comments, statements regarding “spillover events from feral wild boars to the farm settings or even to humans” should also be deleted from abstract (line 31-32).

Response: Thank you for your comment. We have deleted the sentence.

  1. Line 241-242: age information of wild boars has been made available in the revised manuscript. The sentence “age information of wild boars was not available for further interpretation of HEV serology” should be removed.

Response: We have deleted the sentence accordingly.

  1. The authors’ response to item 3 in my previous comment (about significant sampling vias) should be described in the Discussion section.

Response:

Response: We added the response to the Discussion section (line 238-243).

“However, HEV RNA positivity from fecal samples allows a fair comparison between feral and farmed wild boars, and similar age distribution can be observed, with a high-er HEV RNA positivity among 7-12 month feral wild boars (2.3%) and 7-9 month farmed wild boars (4.0%) respectively (Table S1).”

  1. The authors found significantly higher seroprevalence of HEV in farmed wild boars than in feral wild boars (Table 3). However, there were no statistically significant differences between feral wild boars and farmed wild boars in relation to the overall and age-specific prevalence of fecal HEV RNA in the age groups of 0-6, 7-12, and >12 months (Supplementary Table S1). The observed different tendency of the seroprevalence and positivity of fecal HEV RNA between feral and farmed wild boars should be discussed convincingly to support the authors’ claim that farms could be a setting for amplification of HEV.

Response: Thank you for your comment. We observed that the HEV RNA positivity was higher among farmed wild boars (2.6% versus 2.0%, p = 1.000 for wild boars 12 months or younger), and also significant higher seroprevalence in farmed wild boars (75.1% versus 21.6%, p < 0.001 for wild boars 12 months or younger). The higher HEV positivity among farmed wild boars did not reach statistical significance, as a much larger sample size is needed to detect difference in active infection. However, both observations are consistent and provided evidence that the infection risk are higher in farms. This is now further reported and discussed:

“Comparing wild boars 12 months or younger, farmed wild boars had a higher HEV se-roprevalence (2.6% versus 2.0% p = 1.000), though the difference did not reach statistical significance.”

“We also observed a higher HEV RNA positivity among farmed wild boars 12 months or younger compared to feral wild boars (2.6% versus 2.0%), though the difference did not reach statistical significance. This is in line with the observed higher seroprevalence among farmed wild boars, indicating a higher infection risk in farms.”

“First, there were a limited number of samples where HEV genotype was determined to draw conclusions about differences in HEV RNA positivity between farmed and feral wild boars, and HEV genotype distribution between feral and farmed wild boars.”